# EXPERT-BASED REWARD FUNCTION TRAINING: THE NOVEL METHOD TO TRAIN SEQUENCE GENERATORS

**Joji Toyama, Yusuke Iwasawa, Kotaro Nakayama, & Yutaka Matsuo**
Graduate School of Engineering
The University of Tokyo
Hongo, Tokyo, Japan
`{toyama,iwasawa,nakayama,matsuo}@weblab.t.u-tokyo.ac.jp`

## ABSTRACT

The training methods of sequence generator with a combination of GAN and policy gradient has shown good performance. In this paper, we propose expert-based reward function training: the novel method to train sequence generator. Different from previous studies of sequence generation, expert-based reward function training does not utilize GAN's framework. Still, our model outperforms SeqGAN and a strong baseline, RankGAN.

## 1 INTRODUCTION

Generating sequential data is one of the main areas of research in machine learning. Recently, sequence generator training with a combination of generative adversarial nets (GANs) (Goodfellow et al., 2014) and policy gradient (Sutton et al., 2000) has shown good performances, such as Yu et al. (2017); Lin et al. (2017); Guo et al. (2018). Those studies employ a discriminator which is trained to discriminate between a true sequence and a generated sequence, and a generator is trained with policy gradient by treating the discriminator as the reward function.

In this study, we propose expert-based reward function training. In expert-based reward function training, we train the reward function without the generator's samples. Instead, we prepare a proposal distribution which produces a negative sequence given an expert (or true) sequence. The reward function is trained to discriminate between an expert sequence and a sequence from a proposal distribution. Unlike previous studies (Lin et al., 2017; Guo et al., 2018; Yu et al., 2017), expert-based reward function is not a kind of GAN frameworks. Although GAN framework has an advantage that a reward function is simultaneously trained with a generator's performance, the training of the generator frequently fails because of an instability of the GAN framework. Expert-based reward function training prioritizes executing stable training of the generator over taking an advantage of GAN framework.

We conducted experiments based on synthetic data to investigate the effectiveness of expert-based reward function training. As an evaluation method, we employ oracle negative log likelihood (NLL) in synthetic data. We show that our model outperform SeqGAN. We also show that our model outperforms a strong baseline, RankGAN.

## 2 EXPERT-BASED REWARD FUNCTION TRAINING

SeqGAN employs a GAN framework. Although GAN has shown great success, it has also been reported that its training is difficult (Arjovsky & Bottou, 2017). In expert-based reward function training, reward functions are trained by discriminating between expert data and negative data produced from the proposal distribution. Expert-based reward function training does not use the generator's samples at all; therefore, it is not a kind of GAN frameworks. Figure 1 shows the overall proposed training method.

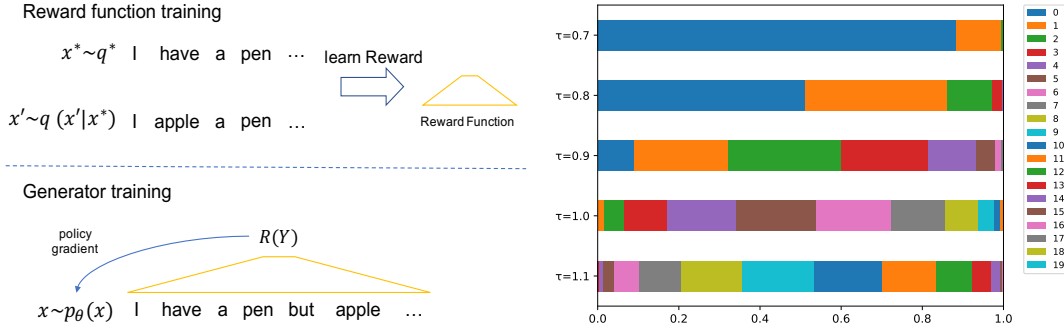

Figure 1: Proposed training method. A reward function is trained to discriminate between a sequence from real data and a sequence from a proposal distribution. Then, a generator is trained with policy gradient.

Figure 2: Fraction of different number of hamming distance applied to a sequence of length 20 and 5,000 vocabulary for different $\tau$. As $\tau$ increases, a proposal distribution produces samples that are farther from experts.

First, we need to prepare the proposal distribution. In this study, we determine the proposal distribution as

$$q(x|x^*) = \frac{1}{Z} \exp\left(\frac{r'(x, x^*)}{\tau}\right) \tag{1}$$

where $r'(x, x^*)$ is a prior reward function. Specifically, we chose negative hamming distance from real data as $r'(x, x^*)$. In this setting, sampling from $q(x|x^*)$ can be easily done as Norouzi et al. (2016) has shown. $\tau$ serves as a hyper-parameter that controls a smoothness of a proposal distribution around the experts. Figure 2 shows the fraction of different numbers of negative hamming distance applied to a sequence of length 20 and 5,000 vocabulary for different values of $\tau$.

Then, we train a reward function $r(x)$ by discriminating between a positive sequence from real data distribution $x^* \sim q^*$ and a sequence from proposal distribution $x' \sim q(x'|x^*)$. $r(x)$ maximizes the following objective

$$L_D = E[\log r(x^*)]_{x^* \sim q^*} + E[\log(1 - r(x'))]_{x' \sim q(x'|x^*)}. \tag{2}$$

After a training converges, we train a sequence generator with policy gradient to maximize the obtained reward function $r(x)$.

In expert-based reward function training, a reward function is trained to discriminate between a sequence from $q^*(x)$ and $q(x)$. It suggests that the density ratio of $q^*(x)$ and $q(x)$ is estimated, and used as a reward. If $q(x)$ is too close to $q^*(x)$, the density ratio is estimated on the samples which are very close to the experts. If $q(x)$ is far different from $q^*(x)$, however, the density ratio around experts are not well estimated. Therefore, it is important to make $q(x)$ produce the samples that are at a proper distance from the experts so that $p_\theta$ can get the good reward signals. The choice of eq.(1) with a negative hamming distance as $r'(x, x^*)$ is suitable in this sense, because we can control a smoothness around an expert by changing $\tau$ as shown in figure 2.

## 3 RELATED WORK

Sequence generative models trained with a combination of a GAN framework and a policy gradient have shown good performance. SeqGAN firstly proposed this training framework(Yu et al., 2017). Some variants of SeqGAN are proposed recently. Lin et al. (2017) proposed RankGAN, in which an original binary classifier discriminator is replaced with a ranking model by taking a softmax over the expected cosine distances from the generated sequences to the real data. Guo et al. (2018) proposed LeakGAN. In LeakGAN utilizes the techniques of hierarchical reinforcement leaning. A discriminator extracts information of uncompleted sentence, and given its information, a manager provides sub-goal to a worker. Different from those methods, expert-based reward function training is not a kind of GAN frameworks.

Table 1: The result of oracle test. PG denotes if the model is trained with policy gradient or not. The top is the model trained with MLE. The score in parentheses is originally reported score in Yu et al. (2017). $\tau = 1.0$ gave the best score in our model.

| Model Name | PG | Adversarial or Expert-based | Oracle NLL |
|---|---|---|---|
| MLE | N | - | 9.03 |
| RAML | N | - | 9.00 |
| SeqGAN | Y | Adversarial | 8.61 (8.73) |
| RankGAN | Y | Adversarial | 8.25 |
| LeakGAN | Y | Adversarial | 7.04 |
| Our model | Y | Expert-based | 7.55 |

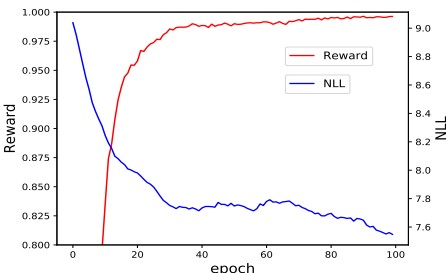

Figure 3: Plots of reward and NLL during the generator's training. We can see that the generator is properly optimized w.r.t. the reward function, and as the returned reward increases, oracle NLL decreases.

Norouzi et al. (2016) proposed reward augmented maximum likelihood (RAML). Norouzi et al. (2016) use a sample from eq.(1) as a target sequence of MLE training. Expert-based reward function training is inspired by this study. We use eq.(1) as the proposal distribution from which a negative sequence is sampled, and a reward function is learned by discriminating those negative samples and the true samples.

## 4 EXPERIMENTS

We examine the effect of expert-based reward function training in synthetic data. For synthetic data experiments, we conduct the oracle test, which was also conducted in Yu et al. (2017); Lin et al. (2017); Guo et al. (2018). In oracle test, we prepare the RNN with fixed parameters, called oracle RNN. We use the sequences of length 20 and 5,000 vocabulary from oracle RNN as training data, and we evaluate the performance of a generator by measuring the NLL of generated sequence with oracle RNN. A reward function is composed by CNN of Kim (2014), which is same as SeqGAN's discriminator. Other than the training method, our model is same as SeqGAN.

Table 1 presents the result of the oracle test. Note that the top is the model trained with only maximum likelihood estimation (MLE). Our model outperforms SeqGAN, indicating that expert-based reward function training is effective. This experimental result shows that the instability of adversarial training causes serious damage to the training of the generator. Our model also outperforms a strong baseline, RankGAN.

In expert-based reward training, the reward function is fixed during the training of the generator, so we can visualize a return of the reward function to see if the policy gradient successfully works. As we can see in figure 3, the generator is properly optimized w.r.t. the reward function, and NLL decreases as a returned reward increases, indicating that the learned reward function is proper and easy to optimize for the generator. It is also noteworthy that $\tau = 1.0$ in eq.(1) gave the best score in our model, in which the proposal distribution provides negative samples that are at proper distance from experts, as shown in figure 2.

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
