# OpenReview forum: "Expert-based reward function training: the novel method to train sequence generators"
_ICLR.cc/2018/Workshop — Accept_

### Official Review · AnonReviewer2 · 2018-03-07
**Training sequence generators using rewards from a (relative) density estimator**

**Rating:** 5
**Confidence:** 4

**Review:**

This paper proposes training a discrete sequence generator, using whatever RL techniques one is comfortable with, to maximize a reward function provided by an "expert" that is separately trained to model the general "shape" of the target distribution. The two obvious difficulties one faces in this direction are (i) how to make the RL-based training work reasonably well (given a fixed reward to optimize), and (ii) how to define an expert that provides useful rewards and is easy to train.

The authors focus on problem (ii). The proposed method is to train a classifier to discriminate between the target distribution and a constructed "contrastive" distribution. The trained classifier can be used as a (relative) density estimate for the target distribution, which can be used as the reward to maximize. I would recommend the authors review the literature on "Noise Contrastive Estimation" and other approaches to unnormalized density estimation.

It's not clear that the proposed method would work better than some obvious baselines, e.g. training an LSTM on the target distribution and then training the generator, via RL, to generate samples that have high log likelihood according to the trained LSTM. This seems kind of like "cheating", since it kind of directly optimizes the reported performance metric, but I'm generally skeptical about metrics that only work when people don't (directly) optimize them.

---

### Official Review · AnonReviewer3 · 2018-03-11

**Rating:** 6
**Confidence:** 4

**Review:**

The paper proposed a novel method for training discrete sequences. It is related to GAN training. The proposed method first trains the discriminator to discriminator real samples and negative samples from data, it then trains the generator to maximize the reward of the discriminator while keeping the discriminator fixed.  The proposed method outperforms SeqGAN and RankGAN for the Oracel test.

This is an interesting idea. It seems that this method alleviates the stability issue of training GANs. For the experiments, was the generator pretrained on the MLE objective.  I would be curious to see the performance on real text data (compared to synthetic data), I am also curious if the generator has issues with mode dropping.

---

### Official Review · AnonReviewer1 · 2018-03-12
**Expert-based reward sequence training review**

**Rating:** 7
**Confidence:** 3

**Review:**

A good set of promising results on a model that essentially uses an intermediate density model to train sequences. This I think is a good way to ensure that the discriminator is measuring something sensible (as the generated distribution is usually likely very disjoint from the true distribution). Is there some sense in doing some curriculum learning here, mixing in the generated samples as it improves?

---

### Decision · Program_Chairs · 2018-03-20
**ICLR 2018 Workshop Acceptance Decision**

**Decision:**

Accept

**Comment:**

Congratulations, your paper was accepted to the ICLR workshop.